# Hyperthermic Intrathoracic Chemotherapy (HITOC) after Cytoreductive Surgery for Pleural Malignancies—A Retrospective, Multicentre Study

**DOI:** 10.3390/cancers13184580

**Published:** 2021-09-12

**Authors:** Michael Ried, Julia Kovács, Till Markowiak, Karolina Müller, Gunnar Huppertz, Michael Koller, Hauke Winter, Laura V. Klotz, Rudolf Hatz, Julia Zimmermann, Bernward Passlick, Severin Schmid, Mohamed Hassan, Martin E. Eichhorn, Hans-Stefan Hofmann

**Affiliations:** 1Department of Thoracic Surgery, University Hospital Regensburg, 93053 Regensburg, Germany; till.markowiak@ukr.de (T.M.); hans-stefan.hofmann@ukr.de (H.-S.H.); 2Department of Thoracic Surgery, Ludwig-Maximilians-University of Munich, Asklepios Lung Clinic, 82131 Gauting, Germany; julia.Kovacs@med.uni-muenchen.de (J.K.); rudolf.hatz@med.uni-muenchen.de (R.H.); julia.zimmermann@med.uni-muenchen.de (J.Z.); 3Center for Clinical Studies, University Hospital Regensburg, 93053 Regensburg, Germany; karolina.mueller@ukr.de (K.M.); gunnar.huppertz@ukr.de (G.H.); michael.koller@ukr.de (M.K.); 4Department of Thoracic Surgery, Thoraxklinik, University Hospital Heidelberg, 69126 Heidelberg, Germany; hauke.winter@med.uni-heidelberg.de (H.W.); laura.klotz@med.uni-heidelberg.de (L.V.K.); martin.eichhorn@med.uni-heidelberg.de (M.E.E.); 5Translational Lung Research Center (TLRC) Heidelberg, Member of the German Center for Lung Research (DZL), 69120 Heidelberg, Germany; 6Department of Thoracic Surgery, Medical Center—University of Freiburg, 79106 Freiburg im Breisgau, Germany; bernward.passlick@uniklinik-freiburg.de (B.P.); severin.schmid@uniklinik-freiburg.de (S.S.); mohamed.hassan@uniklinik-freiburg.de (M.H.); 7Faculty of Medicine, University of Freiburg, 79110 Freiburg im Breisgau, Germany; 8Department of Thoracic Surgery, Hospital Barmherzige Brüder Regensburg, 93049 Regensburg, Germany

**Keywords:** HITOC, hyperthermic intrathoracic chemotherapy, chemoperfusion, pleural malignancy, pleural mesothelioma, cytoreductive surgery

## Abstract

**Simple Summary:**

There continues to be little research in the literature on perioperative outcomes after cytoreductive surgery (CRS) combined with intraoperative hyperthermic chemotherapy-lavage (HITOC) in patients with malignant pleural tumours. The aim of this multicentre study was to assess the results of the current practice in Germany so as to give recommendations to standardize the procedure. CRS with cisplatin-based HITOC can be performed with low major morbidity and a low rate of renal insufficiency, which was associated with the cisplatin dosage of irrigation.

**Abstract:**

In the context of quality assurance, the objectives were to describe the surgical treatment and postoperative morbidity (particularly renal insufficiency). A retrospective, multicentre study of patients who underwent cytoreductive surgery (CRS) with cisplatin-based HITOC was performed. The study was funded by the Deutsche Forschungsgemeinschaft (DFG, German Research Foundation (GZ: RI 2905/3-1)). Patients (*n* = 350) with malignant pleural mesothelioma (*n* = 261; 75%) and thymic tumours with pleural spread (*n* = 58; 17%) or pleural metastases (*n* = 31; 9%) were analyzed. CRS was accomplished by pleurectomy/decortication (P/D: *n* = 77; 22%), extended P/D (eP/D: *n* = 263; 75%) or extrapleural pneumonectomy (EPP: *n* = 10; 3%). Patients received cisplatin alone (*n* = 212; 61%) or cisplatin plus doxorubicin (*n* = 138; 39%). Low-dose cisplatin (≤125 mg/m^2^ BSA) was given in 67% of patients (*n* = 234), and high-dose cisplatin (>125 mg/m^2^ BSA) was given in 33% of patients (*n* = 116). Postoperative renal insufficiency appeared in 12% of the patients (*n* = 41), and 1.4% (*n* = 5) required temporary dialysis. Surgical revision was necessary in 51 patients (15%). In-hospital mortality was 3.7% (*n* = 13). Patients receiving high-dose cisplatin were 2.7 times more likely to suffer from renal insufficiency than patients receiving low-dose cisplatin (*p* = 0.006). The risk for postoperative renal failure is dependent on the intrathoracic cisplatin dosage but was within an acceptable range.

## 1. Introduction

Cytoreductive surgery in combination with hyperthermic intrathoracic chemotherapy (CRS-HITOC) has been reported as a promising surgical therapy for selected patients with pleural malignancies [1]. In recent literature, data regarding this combination therapy within a multimodal treatment approach are mainly available for patients with malignant pleural mesothelioma (MPM) and thymic tumours with pleural dissemination (stage IVa) and rarely for highly selected patients with secondary pleural carcinosis [2,3,4]. The spreading growth pattern of these malignant pleural tumours along the pleura impairs microscopically complete resection, and small residues of the tumour are suspected to foster local recurrence with a negative impact on survival. The concept of CRS has been developed in MPM surgery and can be adapted for appropriate patients with secondary pleural tumours [5]. Macroscopic complete tumour resection (MCR) is recommended whenever possible and is a prerequisite for a favourable prognosis in the case of a curative therapeutic approach [6,7,8,9]. After successful CRS, intraoperative hyperthermic perfusion of the thoracic cavity with chemotherapeutic agent(s) (HITOC) is expected to obtain better local tumour control and thereby improve progression-free and overall survival [10,11,12,13]. So far, there are many unanswered questions regarding the addition of intraoperative HITOC. There is no standardization for HITOC as a procedure or international guideline concerning the administration of chemotherapeutic agents, optimal duration of perfusion or optimal perioperative management of these patients. Therefore, CRS-HITOC is performed by adapting international reported experiences and individual protocols [14].

The objectives of the German HITOC study are to provide data on current practice as well to standardize the procedure of CRS-HITOC and to evaluate the combination with special respect to perioperative outcomes, including postoperative renal insufficiency.

## 2. Materials and Methods

### 2.1. Study Design

We performed a retrospective, multicentre study on patients who underwent CRS-HITOC for pleural malignancies in Germany. Data collection was obtained from January to March 2020 at four major university centres for thoracic surgery with a high caseload of CRS-HITOC procedures. The study was funded by the Deutsche Forschungsgemeinschaft (DFG, German Research Foundation (GZ: RI 2905/3-1)). The trial is registered in the German Registry of Clinical Studies (DRKS-ID: DRKS00015012). The approval of the ethics committee of the University of Regensburg (reference number: 18-1119-104) and of the ethics committees of the participating centres was obtained. The study protocol was developed and published before collecting and analyzing the data [14].

### 2.2. Inclusion and Exclusion Criteria

The study protocol specified the following inclusion criteria: malignant pleural tumours and HITOC after CRS in one surgery. Conversely, treatment without HITOC and HITOC without CRS (not following consensus-based recommendations) were exclusion criteria [14]. The choice of chemotherapeutic agents (alone or in combination) and also the dosage of chemotherapeutic agents for the intracavitary administration was determined by each hospital according to its own experience and protocols independently of clinical conditions.

### 2.3. Explorative, Complex Endpoint

In the context of quality assurance, the primary objectives of this study were to describe the surgical treatment regimen, to investigate postoperative morbidity (particularly renal insufficiency as the most reported and feared complication) and to assess in-hospital mortality. Furthermore, the results were intended to provide valid data for HITOC techniques, including cisplatin dosing and perioperative management, to enable standardization of this surgical therapy approach.

### 2.4. Data Collection

The REDCap (Research Electronic Data Capture) system was used for data collection. It provides the capability to perform major data management activities within a consistent, auditable and integrated electronic environment (data security, data entry and data validation). Patient-informed consent was not an eligibility criterion because data were processed in a pseudonymized manner in accordance with the European Union General Data Protection Regulation (EU-GDPR) and Bavarian Hospital Law (BayKrG).

### 2.5. Definition of Variables

Data collection included clinical characteristics and perioperative data with special respect to CRS-HITOC. To standardize the units of cisplatin, mg was transformed into mg/m^2^ BSA by dividing mg by BSA, since mg/m^2^ BSA is the most frequently used unit for intrathoracic lavage of cisplatin. Moreover, the concentration of cisplatin was categorized as low-dose (≤125 mg/m^2^ BSA) vs. high-dose (>125 mg/m^2^ BSA). This categorization was defined based on results from the literature and the cisplatin doses collected in this study. Postoperative renal insufficiency was classified according to the KDIGO classification in (acute kidney injury) AKI stage I (mild), AKI stage II (moderate) and AKI stage III (severe) [15]. Postoperative surgical complications were determined according to the Clavien-Dindo classification [16].

### 2.6. Statistical Analyses

Descriptive analyses were performed using frequency (*n*), percentage (%), mean (m), standard deviation (SD), median (med), interquartile range (IQR) and range (min/max). To examine potential risk factors for in-hospital mortality, univariable binary regression models were calculated including cisplatin dose, chemotherapeutic agents, resection method, induction chemotherapy (no vs. yes), intraoperative complications (no vs. yes), postoperative complications (no vs. yes) and surgical revision (no vs. yes). Due to the small number of deceased patients, no multivariable analyses could be conducted. To examine potential risk factors for postoperative renal insufficiency (AKI stage < I vs. AKI stage ≥ I), multivariable binary regression (enter method) was calculated, including cisplatin dose, chemotherapeutic agents, resection method, induction chemotherapy (no vs. yes), medicamentous cytoprotection (no vs. yes) and perioperative fluid balancing (no vs. yes).

A mixed linear model (MLM; unstructured repeated covariance type) was used to evaluate repeated measures of creatinine preoperatively, on postoperative day two and at discharge between cisplatin dose, chemotherapeutic agents and resection method. MLM uses the full data set by replacing missing values using maximum likelihood estimates. Thus, all patients, even those with missing creatinine values at specific time points, were used for the analyses. However, at least one creatinine value had to be assessed. For regression analyses and MLM, cisplatin concentration was categorized in low-dose vs. high-dose, chemotherapeutic agents in cisplatin alone vs. cisplatin and doxorubicin, and resection method in P/D, eP/D and EPP.

All statistical analyses were conducted using the software package SPSS (version 26). The level of significance was set at *p*_two-sided_ ≤ 0.050 for all tests. No adjustments for multiple testing were made.

## 3. Results

### 3.1. Patient and Pleural Tumour Characteristics

Between the onset of the HITOC program in January 2008 and until December 2019, 359 patients were treated with CRS-HITOC. Nine patients were excluded from analyses due to the following exclusion criteria (multiple criteria could be met): not receiving the chemotherapeutic agent cisplatin (*n* = 1 carboplatin, *n* = 1 oxaliplatin and 5-fluorouracil), receiving the combination of cisplatin and mitomycin (*n* = 1), the missing concentration of cisplatin (*n* = 2), application of HITOC not in the same surgery as CRS (*n* = 3) or resection method other than P/D, eP/D or EPP (*n* = 2). Thus, *n* = 350 patients were available for the present analyses (Table 1).

The majority of patients (74%) were male, and the mean age of all patients was 61.3 ± 12.5 years (range 18 to 82 years). Most patients presented with a Karnofsky index of 80 and 90% (*n* = 290; 83%) and ECOG status 0 or 1 (*n* = 334; 95%). Previous histological tumour confirmation was mostly performed by VATS (video-assisted thoracic surgery; *n* = 184; 81%). The following tumour entities were diagnosed: MPM (*n* = 261; 75%), stage IVa thymic tumours (*n* = 58; 17%) and secondary pleural metastases (*n* = 31, 9.0%). Among all patients with MPM (*n* = 261), the subtypes were epitheloid (*n* = 221; 85%), biphasic (*n* = 35; 13%) and sarcomatoid (*n* = 5; 2%). The MPM-specific tumour staging according to the UICC (8th edition) showed that 125 patients (48%) had stage I disease, 39 patients (15%) had stage II disease, 91 patients (35%) had stage III disease and 6 patients (2%) had stage IV disease. Patients with thymic tumours in stage IVa (*n* = 58) were classified according to the WHO classification into histological subtypes: AB (*n* = 1; 2%), B1 (*n* = 8; 14%), B2 (*n* = 24; 41%), B3 (*n* = 9; 16%) and C (*n* = 15; 26%), and one patient was diagnosed with an atypical carcinoid of the thymus. Patients with secondary pleural metastases (*n* = 31) had various primary tumour entities, which included pleural metastases of lung cancer (*n* = 12), ovarian cancer (*n* = 5) or pseudomyxoma peritonei (*n* = 3).

### 3.2. Intraoperative Data of CRS and HITOC

The median time interval between initial tumour diagnosis and surgery was two months (IQR 1 to 4 months, range 0 to 108 months). The median duration of surgery, including HITOC, was 345 min (IQR 292 to 406 min, range 118 to 720 min). Intraoperative data on CRS-HITOC are listed in Table 2. CRS was performed by P/D (*n* = 77; 22%), eP/D (*n* = 263; 75%) or EPP (*n* = 10; 3%). Additional anatomical lung resections were performed in 24 patients (7%). The intraoperative resection status was macroscopic complete resection (MCR; R0/R1) in 300 patients (86%), whereas 50 patients (14%) had no MCR (R2).

In all patients, HITOC was performed directly after CRS within one operative session with a closed thoracic cavity (100%). The median time for HITOC was 60 min (range 28 to 120 min), with a median maximum perfusion temperature of 42.0 °C (range 40.0 to 44.0 °C). The median perfusion volume was 5000 mL (range 2500 to 7500 mL). Cisplatin alone was administered in 212 patients (61%), and 138 patients (39%) received a combination of cisplatin plus doxorubicin. The median cisplatin dosage was 108.8 mg/m^2^ BSA (IQR 100.0 to 150.0 mg/m^2^, range 51.2 to 200.0 mg/m^2^). The median doxorubicin concentration was 46.9 mg/m^2^ BSA (IQR 35.7 to 52.9 mg/m^2^, range 15.4 to 66.7 mg/m^2^).

HITOC-associated intraoperative complications (*n* = 6; 2%) included haemodynamic instability and/or relevant parenchymal leakage during perfusion, which made a reduction of the pump-flow (*n* = 5) or stopping of the perfusion (*n* = 1) necessary.

### 3.3. Postoperative Treatment Complications

Postoperative treatment complications (*n* = 178; 51%) are presented in Table 3. In most cases, complications were categorized as grade II (*n* = 61) or grade III (*n* = 96) according to the Clavien-Dindo classification. In 51 patients (15%), surgical revision was performed for various reasons, including parenchymal air leakage (*n* = 13) and hematothorax (*n* = 13). The majority of patients were extubated directly after surgery (*n* = 312; 89%), whereas 15 patients (4%) required prolonged ventilation >24 h postoperatively. Postoperative pneumonia (15%), prolonged parenchymal fistula (air leak > 7 days; 11%), new atrial fibrillation (12%) and respiratory insufficiency (10%) were the most frequently documented nonsurgical complications. Return to the intensive care unit (ICU) was necessary for 24 patients (7%). The median stay in the ICU was two days (IQR 1 to 4 days, range 0 to 56 days), and the median hospital stay was 18 days (IQR 15 to 28 days, range 3 to 139 days). In-hospital mortality was 3.7% (*n* = 13) and verifiable to surgical complications in six patients as well as to nonsurgical complications in seven patients.

In total, 41 (11.7%) patients suffered from postoperative renal insufficiency. Of these 41 patients, 23 had stage I AKI, 10 had stage II AKI and 8 had stage III AKI. The distribution of AKI stages was separately presented for cisplatin dose (Figure 1A) and treatment with cisplatin alone or in combination with doxorubicin (Figure 1B). Five patients (1.4%) required temporary dialysis after surgery (all AKI stage III).

#### 3.3.1. Risk Factors for In-Hospital Mortality

The probability of in-hospital mortality was 12.2 times higher (95% CI = 2.6/55.8) in patients treated with high-dose cisplatin than in patients treated with low-dose cisplatin (*p* = 0.001; Table 4). The probability of in-hospital mortality was 8.6 times higher (95% CI = 1.6/46.1) in patients with intraoperative complications than in patients without intraoperative complications (*p* = 0.012). The combination of cisplatin with doxorubicin, resection method, induction chemotherapy, intraoperative complications, surgical revision and perfusion volume had no impact on in-hospital mortality (*p*-values > 0.050). Due to quasi-separated data, no binary regression could be performed for postoperative complications.

#### 3.3.2. Risk Factors for Renal Insufficiency

In total, 350 patients were included in the regression model (Table 5): 41 patients with renal insufficiency and 309 patients without renal insufficiency. Patients receiving a high dose of cisplatin were 2.7 times (95% CI = 1.3/5.6) more likely to suffer from renal insufficiency than patients receiving a low dose of cisplatin (*p* = 0.006). Patients receiving medicamentous cytoprotection were 8.3 times (95% CI = 2.4/29.5) more likely to suffer from renal insufficiency than patients not receiving cytoprotection (*p* = 0.001). Patients without perioperative fluid balancing were 6.3 times (95% CI = 1.6/25.5) more likely to suffer from renal insufficiency (*p* = 0.009). The combination of cisplatin plus doxorubicin, resection method, and induction chemotherapy had no effect on the development of postoperative renal insufficiency (*p*-values > 0.050).

#### 3.3.3. Perioperative Trend of Creatinine Values

The perioperative trend of creatinine values was examined in 330 patients. Patients who died in the hospital (*n* = 13) and patients without any creatinine value (*n* = 7) were excluded from MLM. As clinically expected, the creatinine value changed over time (*p* < 0.001, Figure 2a). After a significant increase in the mean creatinine value from presurgery to two days after surgery (*p* < 0.001), the mean creatinine value decreased until discharge (*p* = 0.001). The course of perioperative creatinine values was similar in each analyzed subgroup.

Creatinine values were significantly higher in patients treated with low-dose cisplatin than in patients treated with high-dose cisplatin presurgery (*p* = 0.001) and at discharge (*p* = 0.001) (Figure 2b). Creatinine values were significantly higher in patients treated with cisplatin in combination with doxorubicin than in patients treated with cisplatin alone two days after surgery (*p* = 0.023) (Figure 2c). No further significant differences were found.

## 4. Discussion

The results of this retrospective, multicentre study show the implementation and performance of CRS-HITOC in four German centres over the last 12 years. Our results confirmed that CRS-HITOC is feasible and can be performed with low rates of major morbidity and low mortality. The overall rate of postoperative renal insufficiency was 12% due to a slight, temporary increase in the postoperative creatinine values with little clinical impact. The main indication for CRS-HITOC was MPM (75%) with mainly epitheloid subtype (85%) and operable UICC stages I-III (98%). Most data are available for HITOC in patients with MPM, and phase I-II studies have suggested cisplatin as the first choice intrathoracic chemotherapeutic agent [10,11,17,18]. The second indication in our study population was stage IVa thymic tumours with pleural spread (17%). For this tumour entity, some retrospective single-centre studies with small sample sizes showed encouraging survival rates [3,12]. The last indication comprised patients with secondary pleural metastasis (8%) from various primary malignancies. For patients with secondary pleural carcinosis, the indication for CRS-HITOC therapy should be carefully evaluated, and decisions should only be made on a case-by-case basis due to the lack of a large cohort analysis concerning the benefit on overall survival [4]. For patients not eligible for the procedure, palliative treatment should be scheduled.

Following recommendations in the literature, the vast majority of our patients underwent lung-sparing surgery (P/D or eP/D) [13,19]. Since EPP represents a considerably more extensive intervention, patients after EPP and HITOC should be considered separately regarding surgery- or HITOC-related complications [2,20]. A high rate of MCR after CRS is crucial for patient prognosis and builds the basis for additional HITOC since HITOC is expected to only reach microscopic residual tumour cells in the thoracic cavity [1,10,21,22]. It is not yet clear to what extent the effects of HITOC are effective if the residues are larger than a few millimetres (R2).

International studies with HITOC regarding the perioperative outcome and renal insufficiency in patients with MPM demonstrated that higher concentrations of cisplatin lead to an increased risk of postoperative renal insufficiency up to 57%, for which reason the maximum tolerable dose of cisplatin was identified at 225 mg/m^2^ BSA [2,23,24]. In our study, the calculated median cisplatin concentration was 109 mg/m^2^ BSA for all patients with a minimum dose of 51 mg/m^2^ BSA and a maximum dose of 200 mg/m^2^ BSA. Approximately 67% of patients received low-dose cisplatin (≤125 mg/m^2^ BSA), and 33% received high-dose cisplatin (>125 mg/m^2^ BSA). Together, most of our study patients received lower cisplatin dosages compared to studies on HITOC after CRS [10,11,13]. In particular, there was a development over time towards higher cisplatin dosages due to increasing clinical research within the study centres and due to the increasing number of data in the literature. Postoperative AKI was within an acceptable range in our study and was mainly associated with mild and moderate AKI stages (only temporary increase in perioperative creatinine values) with minor clinical impact. It is undisputed that the postoperative creatinine increase and consequent postoperative AKI were also caused in particular by major surgical intervention (CRS) and not only by HITOC. Although the type of CRS had no effect, our data confirmed that patients with EPP had significantly higher postoperative creatinine values than patients with lung-sparing P/D or eP/D. Our data also showed the dose-dependent risk for postoperative renal insufficiency, with higher doses of cisplatin resulting in an almost three times higher likelihood of developing renal insufficiency. However, we would substantiate the available recommendations of an intrathoracic dosing of cisplatin (range 125–175 mg/m^2^ BSA) since renal complications were rare and mild in the majority of patients [2,10,11,24,25]. In addition, the potential and generally temporary adverse effects on renal function must be weighed against the survival benefit of higher cisplatin dosages suggested in the literature [13,26]. The effects of renal protective agents (e.g., amifostine and sodium thiosulfate) in addition to perioperative fluid balancing might be beneficial [2,11,20,24,25,26]. Unfortunately, our data were not uniform regarding additional cytoprotective agents given in each participating centre; therefore, no clear recommendation can be expressed yet. Our data showed a discrete increase in postoperative serum creatinine in the patients with medicamentous cytoprotection, but this had no clinical impact. So far, the use of perioperative fluid balancing is standard and had a positive impact on postoperative renal function, so we can recommend it based on our data.

The mechanism of cisplatin-induced nephrotoxicity after systemic, intravenous administration is complex and is a well-known side effect in approximately one-third of patients. Renal complications defined as AKI with the need for renal replacement therapy occurred in 2.7% (9 of 328 patients) of patients after EPP without HITOC [27]. In a recent observational study of 503 patients with MPM who underwent CRS (43% EPP, 57% pleurectomy) with high-dose intraoperative cisplatin (*n* = 412; 175 to 225 mg/m^2^ cisplatin and/or 910 mg/m^2^ gemcitabine) or without intrathoracic lavage (*n* = 91), 48% of patients developed some degree of AKI. Nearly 54% of patients who received intraoperative cisplatin developed AKI, including severe AKI with a requirement for renal replacement therapy in 3.2% of patients. These patients also had a longer length of stay (26 vs. 13 days) and a 2.7-fold increased risk of death [26]. Nevertheless, 24% of the patients who did not receive intraoperative cisplatin lavage also developed some degree of AKI. Compared to our obviously lower overall rate of postoperative AKI (12%; grade III 2.3%) reported, including only 1.4% of patients with temporary renal replacement therapy after HITOC, there seem to be various additional risk factors for intracavitary cisplatin lavage, which should be taken into account (e.g., postoperative complications, surgical revision). The other complications as well as the in-hospital mortality corresponded to the expected risk according to the CRS.

The application of a second chemotherapeutic agent is also heterogeneous, and its impact on morbidity and overall survival has not yet been clarified [11,14,18,26]. Approximately 40% of our study patients received cisplatin plus doxorubicin. Although there was a trend towards a higher incidence of renal insufficiency due to elevated creatinine values in these patients, the combination of both chemotherapeutic agents was not an independent risk factor for acute renal failure and did not raise postoperative morbidity. Nevertheless, potential benefits on patients’ prognosis still have to be demonstrated in upcoming survival analyses of our study population.

Selected subgroups of patients with MPM or other pleural malignancies may benefit from a standardized, surgery-based multimodal approach [8,9,13]. However, recent international guidelines for the treatment of MPM or secondary pleural metastasis neither mention HITOC as a possible part of a multimodality approach nor describe it at all, since adequate data are still lacking [28,29]. In stage IVa thymic tumors, guidelines already mention HITOC as a therapeutic option [30]. In all three situations, additional local therapies are strongly required to improve intrathoracic microscopic disease and local tumour control [31,32]. Previous retrospective studies could indicate potential benefits of additional HITOC, but a clear survival benefit has not yet been demonstrated by prospective studies.

The study has limitations due to its retrospective nature and its analysis of a selected group of patients. CRS-HITOC was performed in all four centres using almost identical approaches that evolved over the documentation period (2008 to 2019). Therefore, the dosages of both chemotherapeutic agents differed between study centres and changed over time. Two clinics increased the cisplatin dosage up to 175 mg/m^2^ BSA, and two clinics used mostly 200 mg cisplatin. Finally, we cannot provide a control group of patients who underwent CRS without HITOC. Our data are valid due to the implementation of a quality-controlled data management system. Long-term data are still lacking. However, data on progression/recurrence-free and overall survival related to the three different tumour entities are already in progress and are expected soon.

## 5. Conclusions

Based on these data with special respect to the technique of CRS-HITOC and its perioperative management, better standardization of this combination therapy is possible as follows: 1. CRS accomplished by lung-sparing (e) P/D; 2. a cisplatin dosage of 125–175 mg/m^2^ BSA is used, 3. combination with doxorubicin is feasible, and 4. perioperative fluid balancing is used.

Most of the postoperative complications were associated with CRS and did not seem to be directly related to additional HITOC. The risk of developing renal insufficiency after CRS-HITOC is dependent on the dosage of cisplatin. Nevertheless, postoperative renal insufficiency appeared as a temporary increase in creatinine values and in most patients without further clinical impairment. All of these issues and caveats need to be addressed in international guidelines and in forthcoming prospective studies.

## Figures and Tables

**Figure 1 cancers-13-04580-f001:**
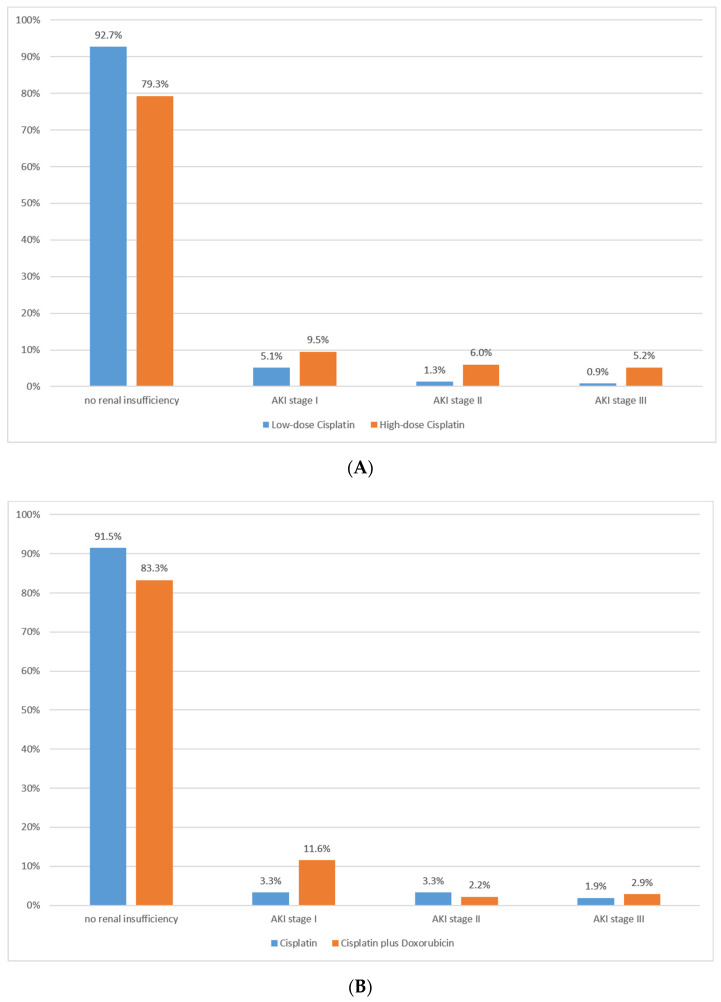
Postoperative renal insufficiency according to AKI stages (**A**) in patients with low-dose (*n* = 234) to high-dose (*n* = 116) cisplatin and (**B**) in patients with cisplatin alone (*n* = 212) compared to cisplatin plus doxorubicin (*n* = 138). AKI = acute kidney injury.

**Figure 2 cancers-13-04580-f002:**
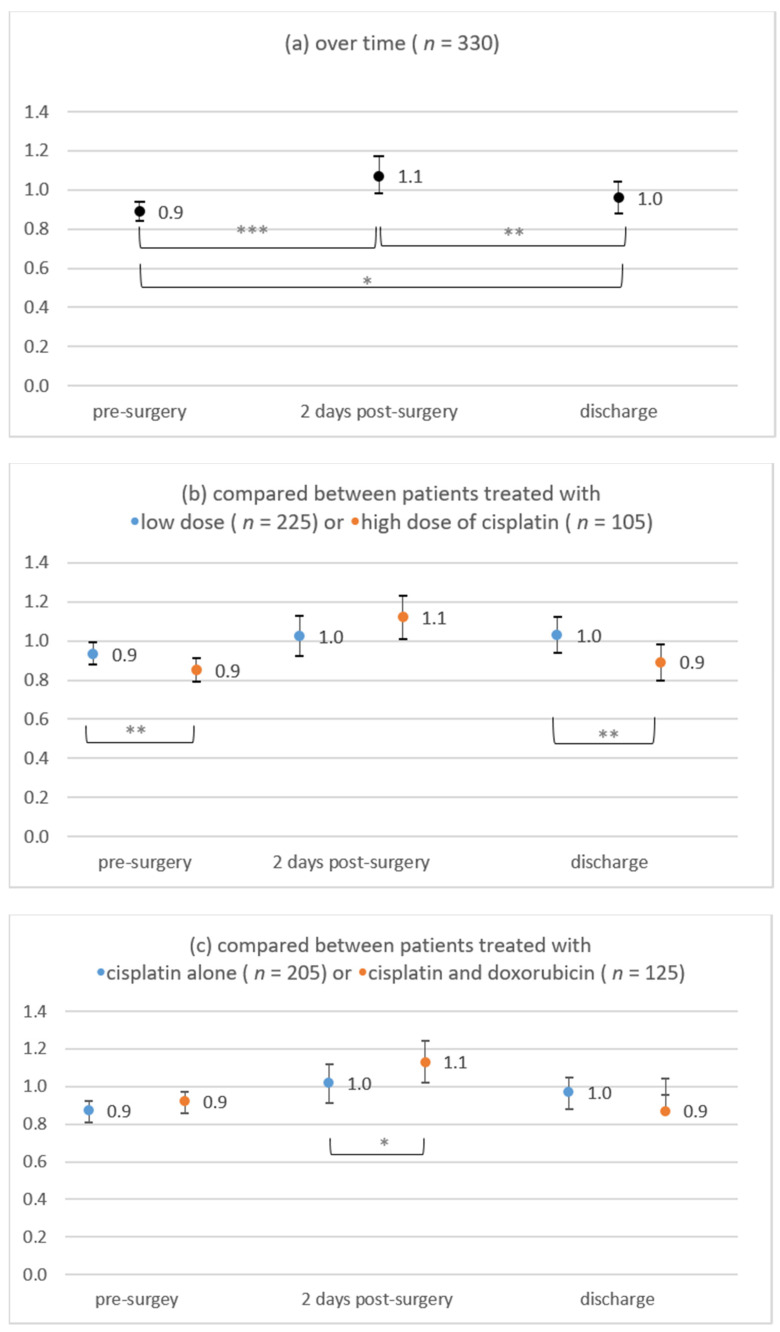
Creatinine values during the perioperative course: (**a**) over time, (**b**) low-dose versus high-dose of cisplatin, (**c**) cisplatin alone versus cisplatin plus doxorubicin, (**d**) depending on surgical cytoreduction. Statistical signifinance differences at * *p* ≤ 0.050, ** *p* ≤ 0.010, and *** *p* < 0.001.

**Table 1 cancers-13-04580-t001:** Patient and pleural tumour characteristics.

Study Population	Study Population
***n* = 350**	***n* (%)**
Sex	
female	92 (26.3)
male	258 (73.7)
Age (mean ± SD) (years)	61.5 ± 12.4
BSA (mean ± SD) (m^2^)	1.93 ± 0.20
BMI (mean ± SD) (kg/m^2^)	26.0 ± 4.1
Karnofsky-index	
60%	1 (0.3)
70%	10 (2.9)
80%	102 (29.1)
90%	188 (53.7)
100%	34 (9.7)
not specified or missing	15 (4.3)
ECOG status	
0	222 (63.4)
1	112 (32.0)
2	1 (0.3)
not specified or missing	15 (4.3)
ASA-classification	
1	22 (6.3)
2	65 (18.6)
3	250 (71.4)
4	5 (1.4)
not specified or missing	8 (2.3)
Confirmation of tumour diagnosis (>100%)	
VATS	284 (81.1)
interventional biopsy	30 (8.6)
prior drainage of pleural effusion	30 (8.6)
not specified or missing	21 (6.0)
Tumour entity malignant pleural mesothelioma thymic tumour stage Iva secondary pleural carcinomatosis or metastases	261(74.6) 58 (16.6) 31 (8.9)
Pleural tumour manifestation	
primary tumour	336 (96.0)
secondary tumour (recurrence)	14 (4.0)
Induction chemotherapy	118 (33.7)

ASA = American Society of Anesthesiologists, BMI = body mass index, BSA = body surface area, ECOG = Eastern Cooperative Oncology Group; SD = standard deviation; VATS = video-assisted thoracic surgery.

**Table 2 cancers-13-04580-t002:** Intraoperative data of CRS-HITOC.

Study Population	Study Population
***n* = 350**	***n* (%)**
Previous thoracic surgery (ipsilateral)	270 (77.1)
Side of surgery	
left	151 (43.1)
right	199 (56.9)
Surgical cytoreduction	
P/D	77 (22.0)
eP/D	263 (75.1)
EPP	10 (2.9)
Resection of	
diaphragm	230 (65.7)
pericardium	156 (44.6)
chest wall	35 (10.0)
lung (wedge/atypical)	133 (38.0)
lung (anatomical)	24 (6.9)
Alloplastic reconstruction/replacement of	
diaphragm	94 (26.9)
pericardium	89 (25.4)
Intraoperative transfusion	179 (51.1)
Intraoperative complication	9 (2.6)
Weaning and extubation in the OR	312 (89.1)
Resection status	
R0/R1 (macroscopic complete)	300 (85.7)
R2 (macroscopic residual disease)	50 (14.3)
Duration of HITOC	
(median, IQR) (min)	60 (60–88)
Temperature perfusate (°C)	
minimum (median, IQR) (n = 313)	41.5 (41.5–42.0)
maximum (median, IQR) (n = 323)	42.0 (42.0–42.5)
Pump flow (mL/min)	
minimum (median, IQR) (n = 308)	1000 (1000–1300)
maximum (median, IQR) (n = 307)	1100 (1000–1500)
Perfusion volume (mL)	
(median, IQR) (n = 339)	5000 (4500–5000)
Chemotherapeutic agents	
cisplatin	212 (60.6)
cisplatin + doxorubicin	138 (39.4)
Calculated dosages of both chemotherapeutic agents (mg/m^2^ BSA)	
cisplatin (n = 350) (median, IQR)	108.8 (100–150)
doxorubicin (n = 138) (median, IQR)	46.9 (35.7–52.9)
Cisplatin concentration (mg/m^2^ BSA)	
low-dose ≤ 125	234 (66.9)
high-dose > 125	116 (33.1)
Medicamentous cytoprotection	178 (50.9)
Fluid balancing	316 (90.3)
Complications during HITOC	6 (1.7)

BSA = body surface area; IQR = interquartile range; P/D = pleurectomy/decortication; eP/D = extended pleurectomy/decortication; EPP = extrapleural pneumonectomy, OR = operating room.

**Table 3 cancers-13-04580-t003:** Postoperative treatment data.

Study Population	Study Population
***n* = 350**	***n* (%)**
Complications	178 (50.9)
Clavien-Dindo classification	
none	172 (49.1)
I	15 (4.3)
II	61 (17.4)
IIIa	35 (10.0)
IIIb	48 (13.7)
IVa	2 (0.6)
IVb	4 (1.1)
V	13 (3.7)
Surgical revision	51 (14.6)
hematothorax	13 (3.7)
pleural empyema	5 (1.4)
parenchymal/bronchial fistula	13 (3.7)
chylothorax	8 (2.3)
other	12 (3.4)
Respiratory insufficiency	36 (10.3)
Prolongated ventilation >24 h	15 (4.3)
Prolongated parenchymal fistula (air leak > 7 days)	40 (11.4)
Postoperative new atrial fibrillation	41 (11.7)
Postoperative transfusion	102 (29.1)
Postoperative sepsis	20 (5.7)
Postoperative pulmonary embolism	6 (1.7)
Postoperative pneumonia	52 (14.9)
Surgical site infection	
yes: cutis, subcutis (CDC)	6 (1.7)
yes: cutís, subcutis, fascia, muscle (CDC)	1 (0.3)
yes: visceral cavity (CDC)	2 (0.6)
no	341 (97.4)
Duration of ICU stay (days) (median, IQR)	2 (1–4)
Return to ICU	24 (6.9)
Duration of hospitalization (days) (median, IQR)	18 (15–28)
In-hospital mortality	13 (3.7)
Renal insufficiency	
AKI stage	41 (11.7)
none	309 (88.3)
I	23 (6.6)
II	10 (2.9)
III	8 (2.3)
Dialysis	5 (1.4)

CDC = Centers for Disease Control and Prevention; ICU = intensive care unit.

**Table 4 cancers-13-04580-t004:** Risk factors for in-hospital mortality.

	In-Hospital Mortality					
	No *n* = 337	Yes *n* = 13	OR	95% CI		*p*
	*n* (%)	*n* (%)				
Cisplatin dose						
low-dose ^1^	232 (99.1)	2 (0.9)				
high-dose	105 (90.5)	11 (9.5)	12.15	2.65	55.80	0.001
Chemotherapeutic agents cisplatin cisplatin and doxorubicin ^1^	205 (96.7) 132 (95.7)	7 (3.3) 6 (4.3)	0.75	0.25	2.28	0.614
Resection method P/D eP/D EPP ^1^	75 (97.4) 253 (96.2) 9 (90.0)	2 (2.6) 10 (3.8) 1 (10.0)	0.24 0.36	0.02 0.04	2.92 3.09	0.263 0.348
Induction chemotherapy						
no ^1^	221 (96.1)	9 (3.9)				
yes	116 (96.7)	4 (3.3)	0.85	0.26	2.81	0.786
Intraoperative complications						
no 1	330 (96.8)	11 (3.2)				
yes	7 (77.8)	2 (22.2)	8.57	1.59	46.10	0.012
Surgical revision						
no ^1^	290 (97.0)	9 (3.0)				
yes	47 (92.2)	4 (7.8)	2.74	0.81	9.27	0.104

The results of univariable binary logistic regression analyses and descriptive distribution of risk factors for in-hospital mortality are presented. ^1^ reference category. Postoperative complications could not be analyzed by means of a binary logistic regression model due to quasi-separated data (i.e., the outcomes can be classified perfectly for some subset of the data).

**Table 5 cancers-13-04580-t005:** Risk factors for renal insufficiency.

	Renal Insufficiency					
	No *n* = 309	Yes *n* = 41	OR	95% CI		*p*
	*n* (%)	*n* (%)				
Cisplatin dose						
low-dose ^1^	217 (92.7)	17 (7.3)				
high-dose	92 (79.3)	24 (20.7)	2.73	1.33	5.63	0.006
Chemotherapeutic agents						
cisplatin ^1^	194 (91.5)	18 (8.5)				
cisplatin and doxorubicin	115 (83.3)	23 (16.7)	0.67	0.26	1.71	0.402
Resection method P/D eP/D EPP ^1^	69 (89.6) 233 (88.6) 7 (70.0)	8 (10.4) 30 (11.4) 3 (30.0)	0.46 0.85	0.09 0.19	2.25 3.73	0.335 0.827
Induction chemotherapy						
no ^1^	207 (90.0)	23 (10.0)				
yes	102 (85.0)	18 (15.0)	1.04	0.50	2.19	0.913
Cytoprotection						
no ^1^	162 (94.2)	10 (5.8)				
yes	147 (82.6)	31 (17.4)	8.36	2.37	29.49	0.001
Perioperative fluid balancing						
no	28 (82.4)	6 (17.6)				
yes ^1^	281 (88.9)	35 (11.1)	6.33	1.57	25.45	0.009

The results of multivariable binary logistic regression analysis (Χ^2^ = 31.3, *p* < 0.001, Nagelkerkes R^2^ = 0.166) and descriptive distribution of risk factors for renal insufficiency are presented. ^1^ reference category.

## Data Availability

Data are available on request due to privacy and ethical restrictions.

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
