# Peer review of "Hyperthermic Intrathoracic Chemotherapy (HITOC) after Cytoreductive Surgery for Pleural Malignancies—A Retrospective, Multicentre Study"

_cancers, 2021, doi:10.3390/cancers13184580_

Round 1
Reviewer 1 Report
This retrospective study on the use of cytoreductive surgery and HITOC in patients with pleural malignancies aims to provide additional retrospective data on the procedure in a multicentre setting. A study protocol has been published before and the analysis was performed accordingly with a focus on perioperative complications.
Several issues should contribute to improve the manuscript.
- While some retrospective series indicate a potential benefit of HITOC and additional well structured multicentre data are beneficial no prospective data to support an advantage of HITOC in any of the reported indications is available. This should be clearly acknowledged.
- The current publication manuscript does not add any information on the actual value of HITOC due to the lack of long-term data. This should be clearly stated in the limitations section
- The patient collective is highly heterogenous. Particularly CRS for pleural spread of extrathoracic tumors is not an established indication. Provide a rationale why these patients were included
- A clear analysis of pre-operative creatinine values and whether this introduced a selection bias into dosage and type of chemotherapy used for HITOC is lacking.
- Information on neoadjuvant chemotherapeutic treatment and its impact on preoperative renal function and postoperative renal insufficiency is missing.
- Why were patients receiving medicamentous cytoprotection 8.3 times more likely to suffer renal insufficiency? Provide an explanation and the implications of this finding on further patient management.
- The dose of ciplatin and the perfusate volume are highly variable. Is any information available on the actual cisplatin concentration reached in the perfusate?
- The study design describes the four participating centers as the ones with the "highest" case-load of HITOC. Provide a source this is based on or rather state "high-volume centers".
Author Response
Several issues should contribute to improve the manuscript.
Comment 1: While some retrospective series indicate a potential benefit of HITOC and additional well structured multicentre data are beneficial no prospective data to support an advantage of HITOC in any of the reported indications is available. This should be clearly acknowledged.
Answer 1: We thank Reviewer 1 for this comment and agree with it. Previous retrospective studies could indicate potential benefits of additional HITOC, but a clear survival benefit has not yet been demonstrated by prospective studies. Therefore, we are looking forward to our further survival data, which will be published next year. Based on our retrospective study results, hopefully prospective studies can also be initiated.
Changes 1: Page 14, lines 361-363.
Comment 2: The current publication manuscript does not add any information on the actual value of HITOC due to the lack of long-term data. This should be clearly stated in the limitations section
Answer 2: We have added this aspect to the limitations section. Survival data are already in progress.
Changes 2: Page 14, line 371.
Comment 3: The patient collective is highly heterogenous. Particularly CRS for pleural spread of extrathoracic tumors is not an established indication. Provide a rationale why these patients were included.
Answer 3: The main indications for CRS with HITOC are malignant pleural mesothelioma and thymomas with pleural metastasis. This describes the literature and could also be confirmed by our data. The smallest subgroup in our study (n= 31) had pleural metastasis from another primary tumour as indication. This represents the indication for surgery with HITOC only in selected patients. These patients were included in this data analysis because the primary purpose was to investigate the technique of CRS with HITOC and its complication spectrum for quality control purposes. The effects on survival of this special subgroup will be analyzed in parallel.
Changes 3: No changes.
Comment 4: A clear analysis of pre-operative creatinine values and whether this introduced a selection bias into dosage and type of chemotherapy used for HITOC is lacking
Answer 4: The level of preoperative serum creatinine does not clinically influence the choice of chemotherapeutic agents or their dosage. The choice was primarily made by the clinics themselves. Our findings even showed that creatinine values were significantly higher in patients treated with low-dose cisplatin than in patients treated with high-dose cisplatin presurgery (figure 2B). Clinically, the difference was not relevant.
Changes 4: Page 3, lines 87-90.
Comment 5: Information on neoadjuvant chemotherapeutic treatment and its impact on preoperative renal function and postoperative renal insufficiency is missing.
Answer 5: Our logistic regression analysis was able to exclude an influence of induction chemotherapy on postoperative kidney function (see table 5). This also confirms my clinical experience, because we had no problems with preoperatively increased creatinine levels or increased postoperative renal problems in any patient.
Changes 5: No changes.
Comment 6: Why were patients receiving medicamentous cytoprotection 8.3 times more likely to suffer renal insufficiency? Provide an explanation and the implications of this finding on further patient management.
Answer 6: Thank you for this important comment. Medicamentous cytoprotection (amifostine and sodium thiosulfate) was performed almost exclusively in patients from our clinic and at the same time high cisplatin doses of 175 mg/m2 BSA were administered in these patients (Markowiak et al. J Surg Oncol 2019; 120 (7): 1220-6). Clinically, the current study showed a discrete increase in postoperative creatinine, but without further clinical relevance. Consequently, although this correlation was shown statistically, it had no further clinical implication to the patient management. We will have to wait for future data in which patients with different doses and uniform medicamentous cytoprotection will be studied. Recent literature (e.g. Hod T, et al. J Thorac Cardiovasc Surg 2021; 161 (4): 1510-1518) examined this currently.
Changes 6: Page 13, lines 323-325.
Comment 7: The study design describes the four participating centers as the ones with the "highest" case-load of HITOC. Provide a source this is based on or rather state "high-volume centers".
Answer 7: In 2018, our working group had made and published a nationwide survey on HITOC in Germany (Ried M, et al. Zentralbl Chir 2018; 143 (3): 301-306). The thoracic surgery clinics with the most HITOC procedures were identified. This was described in more detail in the published study protocol (Markowiak T, et al. BMJ open 2020; 10 (7): e041511). However, we amend the sentence as you have suggested to “high caseloads”.
Changes 7: Page 2, line 76.
Reviewer 2 Report
a very interesting retrospective analysis on outcome of CRS + HITOC of a large cohort of MPM patients, with some interesting outcome. Analysis and discussion is generally sound and certainly worth publishing, but this reviewer is left with some questions and I think the manuscript will gain in value if you address these.
Among your interesting outcomes are the observations that high dose versus low dose cisplatin is associated with substantially more morbidity. Likewise the observations regarding cisplatin alone and in combination with doxorubicin are of interest.
I do think that you should attempt a more in depth analysis to demonstrate there is no bias in these data, so to demonstrate there is no relationship between disease stage and cisplatin dose or combinatorial use. You do mention differences in strategy among the 4 centres and difference over time regarding dose and combined use with doxorubicin. It would be great if you could quantify these statements as these could ease the mind of the reader regarding potential bias. And you could perform multivariate analysis of known risk factors to demonstrate absence of bias between low and high dose, and to show that low and high dose arms are well balanced.
Author Response
Among your interesting outcomes are the observations that high dose versus low dose cisplatin is associated with substantially more morbidity. Likewise the observations regarding cisplatin alone and in combination with doxorubicin are of interest.
Comment 1: I do think that you should attempt a more in depth analysis to demonstrate there is no bias in these data, so to demonstrate there is no relationship between disease stage and cisplatin dose or combinatorial use.
Answer 1: Thank you very much for this comment. The choice of chemotherapeutic agents (alone or in combination) and also the dosage of chemotherapeutic agents does not depend on the tumour stage, but was determined by each hospital according to its own experience and standards. This is the clinical approach at all four hospitals and these data were retrospectively analysed.
Changes 1: Page 3, lines 87-90.
Comment 2: You do mention differences in strategy among the 4 centres and difference over time regarding dose and combined use with doxorubicin. It would be great if you could quantify these statements as these could ease the mind of the reader regarding potential bias.
Answer 2: For example, in our clinic we started with a cisplatin dose of 100 mg/m2 BSA, regardless of the tumour entity or tumour stage, and increased this to 175 mg/m2 BSA, meanwhile in combination with doxorubicin as standard. Similarly, another clinic had increased the cisplatin dosage with increasing clinical experience. The other two clinics always use 200 mg cisplatin. Since in the literature the concentration mg/m2 BSA is usually used, we have converted all cisplatin doses accordingly (Patients and Methods: definition of variables). In the limitations section, we have mentioned the different approaches. We hope that our results can contribute to a more uniform dosing of intrathoracic chemotherapeutic agents.
Changes 2: Page 14, lines 368-369.
Comment 3: And you could perform multivariate analysis of known risk factors to demonstrate absence of bias between low and high dose, and to show that low and high dose arms are well balanced.
Answer 3: Thank you for this comment. As already described above, the choice of chemotherapeutic agents and also their dosage was determined by each individual clinic. In this retrospective analysis, we were able to differentiate between low-dose and high-dose groups at different cisplatin doses. The categorisation into low-dose and high-dose (threshold 125 mg/m2 BSA) was defined based on results from the literature and the cisplatin doses collected in this study. This resulted in the distribution into these two groups. We used this grouping to allow for better comparability and analysis of the defined endpoints. After all, this was exactly one of the aims of our study, the presentation of the HITOC together with the intrathoracic chemotherapy administration and its possible effects. Regarding your suggestion, for in-hospital mortality (n= 13), multivariable analyses are not possible. For risk factors for renal failure, it would be borderline in terms of the number of variables.
Changes 3: Page 3, lines 110-112.
Reviewer 3 Report
Your article is interesting and well presented
I have some questions:
- According to which criteria were patients treated with cisplatin alone or in combination with doxorubicin? Was it related to the different habits of the centres participating in the study?
-
Was thehigh or low cisplatin dose decided according to the pre-operative creatinine value or were there other criteria (center habits,..)
-
I am not an expert in thoracic surgery but I routinely perform cytoreduction surgery and intraperitoneal chemohyperthermia. I am surprised that the majority of patients were classified as ASA 3 and there were also ASA 4 because this type of surgery is long and sometimes has high morbidity rate
-
Why was sodium thiosulphate or a similar molecule not used routinely for all patients? Usually if patients are well hydrated peroperatively and receive nephroprotection there is no renal failure or dialysis postoperatively
- In your article you talk about R2 resection. Literature has shown thet in abdominal surgery you lose the benefit of chemohyperthermia if the residue is greater than a few millimetres. Isn't it the same in thoracic surgery?
Author Response
Comment 1: According to which criteria were patients treated with cisplatin alone or in combination with doxorubicin? Was it related to the different habits of the centres participating in the study?
Answer 1: Thank you for this comment. The choice of intrathoracic chemotherapeutic agents, alone or in combination, was made by each hospital. In our clinic, we started with cisplatin alone in the first years and then added doxorubicin a few years ago. Since there have been few studies on this topic and no clear recommendations, the chemotherapeutic agents applied vary. So far, only cisplatin has been defined as the chemotherapeutic agent of choice, a combination with doxorubicin and recently also with gemcitabine is described in the literature. Further studies and their results have to be awaited. In the limitations section we have mentioned the different approaches.
Changes 1: Page 3, lines 87-90; page 14, lines 368-369.
Comment 2: Was the high or low cisplatin dose decided according to the pre-operative creatinine value or were there other criteria (center habits,..)
Answer 2: This is an important question, which I must answer similarly to the question before. Here, too, we still lack valid data, so that the dosage of cisplatin was mostly based on the experience or standards of the clinics. Thus, we started with low-dose cisplatin of 100 mg/m2 BSA and were able to increase the dosage to 175 mg/m2 BSA in the absence of complications or adjusted perioperative management. Meanwhile, in the literature, the dosage of 175 mg/m2 BSA is considered safe and therefore also recommended for combination administration. The level of cisplatin dosage is not dependent on the preoperative serum creatinine, but mainly on the standard of the clinic. Our findings even showed that creatinine values were significantly higher in patients treated with low-dose cisplatin than in patients treated with high-dose cisplatin presurgery. Clinically, the difference was not relevant.
However, again, we hope that our results can contribute to a more uniform dosing of intrathoracic chemotherapeutic agents.
Changes 2: Page 3, lines 87-90; page 14, lines 368-369.
Comment 3: I am not an expert in thoracic surgery but I routinely perform cytoreduction surgery and intraperitoneal chemohyperthermia. I am surprised that the majority of patients were classified as ASA 3 and there were also ASA 4 because this type of surgery is long and sometimes has high morbidity rate.
Answer 3: That is correct. Most patients were classified by the anesthesiologists in ASA 3 or 4 due to their underlying malignant disease. Thus, this classification is mainly due to the malignant tumour disease with limited survival prognosis and provides less information about the actual clinical condition of the patient. I think the ECOG status or the Karnofsky index are better parameters to describe the preoperative general condition of the patients. For very large chest surgery, patients must be in clinically good condition and consequently classified as functionally operable.
Changes 3: No changes.
Comment 4: Why was sodium thiosulfate or a similar molecule not used routinely for all patients? Usually if patients are well hydrated peroperatively and receive nephroprotection there is no renal failure or dialysis postoperatively
Answer 4: Unfortunately, there are no standards in the area of additional nephroprotection here either. In 2013, we started the administration of sodium thiosulfate and amifostine in our clinic after good data were published by Prof. Sugarbaker and colleagues. Our results also confirmed the positive effects of additional nephroprotection (Markowiak et al. J Surg Oncol 2019; 120 (7): 1220-6) and in combination with fluid balancing we have almost no renal problems. However, even in this area clear recommendations are lacking, so that not all clinics have applied additional, uniform nephroprotection. Therefore, unfortunately, we cannot provide clear recommendations with these data, so that further analyses must follow in this area. So far, only perioperative fluid balancing is performed as standard, so we can also recommend it based on our data.
Changes 4: Page 13, lines 325-327.
Comment 5: In your article you talk about R2 resection. Literature has shown thet in abdominal surgery you lose the benefit of chemohyperthermia if the residue is greater than a few millimetres. Isn't it the same in thoracic surgery?
Answer 5: This is a very important aspect. The literature describes a worse survival after R2 resection of pleural tumors. To my knowledge, it is not yet clear to what extent the effects of HITOC are effective in this case. Data are still missing. Perhaps our following survival data can shed more light on this aspect.
Changes 5: Page 13, lines 293-294.
Round 2
Reviewer 1 Report
The issues raised have been answered appropriately and implemented in the manuscript.
Reviewer 2 Report
The authors have addressed all my concerns!